# A New Fungal Triterpene from the Fungus *Aspergillus flavus* Stimulates Glucose Uptake without Fat Accumulation

**DOI:** 10.3390/md20030203

**Published:** 2022-03-10

**Authors:** Dan-dan Li, Ying Wang, Eun La Kim, Jongki Hong, Jee H. Jung

**Affiliations:** 1College of Pharmacy, Pusan National University, Busan 46241, Korea; leedany1992@gmail.com (D.-d.L.); wangrunhe829@gmail.com (Y.W.); eunlakim@gmail.com (E.L.K.); 2College of Pharmacy, Kyung Hee University, Seoul 02447, Korea; jhong@khu.ac.kr

**Keywords:** PPAR, triterpenoid, glucose uptake, adipogenesis, antidiabetic, asperflagin, *Aspergillus flavus*

## Abstract

Through activity-guided fractionation, a new triterpene (asperflagin, **1**) was isolated as a PPAR-γ agonist from the jellyfish-derived fungus *Aspergillus flavus*. Asperflagin displayed selective and moderate transactivation effects on PPAR-γ in Ac2F rat liver cells. Based on further biological evaluation and molecular docking analysis, we postulated that asperflagin might function as a PPAR-γ partial agonist. This compound was calculated to display a typical PPAR-γ ligand–receptor interaction that is distinct from that of full agonistic antidiabetics such as rosiglitazone, and may retain the antidiabetic effect without accompanying weight gain. Weight gain and obesity are typical side effects of the PPAR-γ full agonist rosiglitazone, and lead to suboptimal outcomes in diabetic patients. Compared to rosiglitazone, asperflagin showed higher glucose uptake in HepG2 human liver cells at concentrations of 20 and 40 μM but induced markedly lower adipogenesis and lipid accumulation in 3T3-L1 preadipocytes. These results suggest that asperflagin may be utilized for further study on advanced antidiabetic leads.

## 1. Introduction

Owing to their remarkable scaffold diversity, structures of natural products contribute to the discovery of potent leads for drug development [1]. Especially, natural products isolated from marine microorganisms, green algae, sponges, or cnidarians [2] have always been attractive due to their remarkably high hit rates in biological evaluation [3]. Compared to other marine organisms, microorganisms are more amenable to culture. Accordingly, more sustainable supply of bioactive components can be obtained for intensive in vivo and preclinical studies [4].

According to the report of the International Diabetes Federation, worldwide diabetic patients will reach 642 million or more by 2040 [5]. In the clinic, approximately 90% of diabetic patients have type 2 diabetes (T2DM), which is associated with hyperglycemia, glucose intolerance, and insulin resistance [5]. In recent years, peroxisome proliferator-activated receptor (PPAR) ligands, such as fibric acid and thiazolidinediones (TZDs), have frequently been employed to combat T2DM or metabolic syndrome [6]. 

PPARs are ligand-activated nuclear transcription factors that play crucial roles in regulating lipid metabolism, glucose homeostasis, adipogenesis, and inflammation [7,8]. PPARs comprise three subtypes, namely, PPAR-α, -β/δ, and -γ. Especially PPAR-γ was principally recognized as a potential antidiabetic drug target. In the cytosol, PPAR-γ receptor is activated by binding with agonistic ligands, and the activated PPAR-γ is translocated into the nucleus to dimerize with retinoid X receptor (RXR). This PPAR-γ/RXR complex binds to PPAR response elements (PPRE) to induce the expression of genes responsible for antidiabetic effects (glucose uptake, adipogenesis, etc.). Thereby, PPAR-γ agonistic ligands could promote glucose uptake and reduce the blood lipid level through activating PPAR-γ. PPAR-γ is well-known as a potential target that could improve glucose homeostasis by increasing the glucose and insulin sensitivity of tissues [9]. PPAR-γ activation could also reduce the blood lipid level by absorbing and accumulating lipid into fat cells via adipogenesis stimulation.

TZD drugs such as rosiglitazone exert antidiabetic functions mainly by activating PPAR-γ [10]. However, clinically used TZD drugs (PPAR-γ full agonist) have side effects, such as weight gain, fluid retention, and lowering of blood pressure [11], with weight gain being particularly undesirable to diabetic patients. In contrast to PPAR-γ full agonists, several partial agonists were reported to enhance insulin sensitization without triglyceride (TG) accumulation, ultimately evading full agonist-induced side effects, especially weight gain [12]. The synthetic PPAR-γ partial agonist, INT131, displayed equipotency as rosiglitazone in improving glucose tolerance but with less weight gain and lower plasma volume [13]. Additional synthetic PPAR-γ partial agonists include nTZDpa [14], SR145 [14], BVT.13 [14], and metaglidasen [15,16]. A natural PPAR-γ partial agonist, amorfrutin, improved insulin resistance without inducing a concomitant increase in fat storage or other unwanted side effects in diet-induced obese and db/db mice [17]. Therefore, to evade side effects, such as lipid accumulation and body weight gain induced by the PPAR-γ full agonist, PPAR-γ partial agonists have been considered as new potential antidiabetic candidates [10]. Accumulating evidence indicates that the PPAR-γ protein/ligand complex incorporates coregulatory proteins [10]. The different recruiting profile of coregulatory proteins may affect the therapeutic and/or adverse effects of ligands. PPAR-γ full agonists, such as rosiglitazone, directly interact with the helix 12 (H12) and AF-2 surface of PPAR-γ ligand-binding domain (LBD) through H-bonds to Tyr^473^ (H12) and other key amino acid residues, including Ser^289^, His^323^, and His^449^ [14,18]. However, side effect-evading PPAR-γ partial agonists did not show direct interactions with H12. Instead, they stabilized other regions of LBD through H-bonding with Ser^342^ of β-sheet and hydrophobic interactions with the H3 region [14,18]. 

In our search for PPAR-γ agonists, 2,5-diketopiperazines were previously isolated from the jellyfish-derived fungus *Aspergillus flavus* [19]. From the same organism, a new triterpene (**1**) (Figure 1) and an isochromanone derivative (**2**) [20] were isolated as PPAR-γ partial agonists. Compound **1** displayed selective and moderate transactivation effects on PPAR-γ, suggesting its potential as a partial agonist. Docking simulation also suggested that **1** binds to the PPAR-γ LBD in a similar pose as other known partial agonists. In subsequent cell-based biological evaluation, **1** showed higher glucose uptake-stimulating activity than rosiglitazone at higher concentrations (HepG2 cells, 20 and 40 μM), and induced negligible lipid accumulation (3T3-L1 cells). Such findings suggest that **1** may serve as a PPAR-γ partial agonist that retain a therapeutic effect (glucose uptake stimulation) but is less associated with side effects (lipid accumulation) than the typical full agonist, rosiglitazone.

## 2. Results and Discussion

### 2.1. Structure of the New Compound

Compound **1** was obtained as a yellow oil. The molecular formula C_30_H_48_O_5_ was determined using HRESI mass spectral data (*m/z* 533.3484 [M + HCOO]^−^, ∆+ 1.1 ppm), which suggested seven degrees of unsaturation. A comparison of the ^1^H and ^13^C NMR data of **1** (Table 1) with that of 15α,22β,24-trihydroxyolean-11,13-diene-3-one [21] revealed that they share common skeletal features. The olefinic proton signals at δ_H_ 6.36 (dd, H-12) and δ_H_ 5.64 (d, H-11) and the HMBC correlations indicated that the two double bonds were conjugated and located at C11/C12 and C13/C18, similar to the known compound 15α,22β,24-trihydroxyolean-11,13-diene-3-one [21]. In addition to 15,22,24-trihydroxyolean-11,13-diene skeleton, **1** showed hydroxyl substitutions at C-3 and C-30. The key COSY and HMBC correlations (Figure 2A) indicated a hydroxyl substitution at C-3 and C-30. The COSY correlations from δ_H_ 4.03 (H-15) to H-16 (δ_H_ 1.60, 1.91) and the HMBC correlation from CH_3_-27 (δ 14.9) to δ_H_ 4.03 indicated the hydroxyl substitution at C-15. The COSY correlations from δ_H_ 3.44 (H-22) to H-21 (δ_H_ 1.62, 1.36) and the HMBC correlations from CH_3_-28 (δ 19.1) to δ_H_ 3.44 indicated the hydroxyl substitution at C-22. The key HMBC correlations from H-30 (δ 3.28) to C-29 (δ 20.9) and C-19 (δ 32.9), and CH_3_-29 (δ 0.80) to C-19 (δ 32.9) and C-30 (δ 73.7), indicated the attachment of a hydroxyl group at C-30. The chemical shift of H-3 (δ 3.38; the chemical shift of β-oriented H-3 would be higher than 3.6 ppm) and the coupling constant of 14.0 Hz between 2-H_ax_ and 3-H_ax_ indicated an α-orientation of H-3 [22,23], and the NOE correlations (Figure 2B) among H-3 (δ 3.38)/H-5 (δ 0.95)/H-7 (δ 1.63)/CH_3_-27 (δ 1.00)/H-16b (δ 1.62)/H-22 (δ3.45)/CH_3_-29 (δ 0.80) indicated they are located on the same side as H-3. The NOE correlations (Figure 2B) among H-24 (δ 4.09)/CH_3_-25 (δ 0.92)/CH_3_-26 (δ 0.77)/H-15 (δ 4.03)/CH_3_-28 (δ 1.09) indicated they are located on the opposite side to that of H-3. The multiplicities and the coupling constants may be utilized to define the relative configuration. However, many of the proton signals were severely overlapped and it was hard to measure the exact multiplicity and coupling constant. Finally, the chemical structure of **1** was defined as oleana-11,13(18)-diene-3β,15α,22β,24,30-pentol and was given a trivial name—asperflagin. Triterpenoids in fungi are mostly biosynthesized starting from 2,3-oxidosqualene via cyclization of the chair-boat-chair-boat intermediate, whereas the oleanane-type triterpenoids such as asperflagin are biosynthesized via cyclization of the chair-chair-chair-boat intermediate. Though the presence of oleanane type triterpenoids in fungi is uncommon, there are some precedents in marine-derived fungi [21,24,25].

### 2.2. Identification of the Known Compounds

Other compounds isolated from the active fractions (**2**–**11**) were identified by comparison of ^1^H and ^13^C NMR spectral data with those reported. They were identified as 6,8-dihydroxy-3-((1*E*,3*E*)-penta-1,3-dien-1-yl)isochroman-1-one (**2**) [20], 1-(4-hydroxy-2-methoxyphenyl)-1-propanone (**3**), phomapyrone C (**4**) [26], daidzein (**5**) [27], genistein (**6**) [28], (*S*)-(+)-1-(4-hydroxy-2,6-dimethoxy-3,5-dimethylphenyl)-2-methylbutan-1-one (**7**) [29], acetyl-β-carboline (**8**) [30], MK800-62F1 (**9**) [23], sigmoside C (**10**) [31], and prosapogenol (**11**) [32].

### 2.3. PPAR Transactivation by Compound ***1*** and ***2***

T2DM is a disease with a complex pathogenesis [33], and PPARs participate in the pathogenesis of T2DM [33]. PPARs, including PPAR-α, -β/δ, and –γ isotypes, are transcription factors that are activated by ligands and form a heterodimer with retinoic acid-X receptor (RXR). The heterodimer then binds to peroxisome proliferator response elements (PPREs) of DNA and promote the transcription of target genes [34]. By regulating the transcription of target genes, PPARs play an important role in maintaining glucose and lipid homeostasis [34]. In this study, a luciferase assay was used to evaluate the PPAR transactivation effect of the compounds. Prior to the transactivation assay using Ac2F rat liver cells, the cytotoxicity of the compounds to Ac2F cells was measured to establish appropriate concentrations for the luciferase assay. No significant cytotoxicity of all compounds was observed up to 40 μM (Appendix A). Therefore, 10 μM was selected as the concentration for the PPARs transactivation assay. Compared to the standard PPAR-α agonist, WY-14643, 10 μM of compounds **2**, **5**, and **6** induced moderate activation of PPAR-α (Appendix A). None of the compounds showed notable activity toward PPAR-β/δ (Appendix A). Compared to the standard PPAR-γ agonist, rosiglitazone, **6** displayed significant activity, while compounds **1**, **2**, and **5** displayed moderate activity (Appendix A). Daidzein (**5**) and genistein (**6**) were previously reported to activate PPAR-α and PPAR-γ [35] and exert anti-inflammatory activity. However, these compounds also function as agonists of ER-α and β [36]. The ultimate biological activity of daidzein and genistein was induced by the balance of the ER- and PPAR-mediated pathways, [37] and these drugs were unsuitable as PPAR partial agonistic antidiabetic agents. However, **1** and **2** showed moderate transactivation effects on PPAR-α and/or –γ, respectively, which suggest their potential as partial agonists. Therefore, the PPAR agonistic activity of **1** and **2** was further evaluated using various concentrations (Figure 3). In the luciferase assay, the PPRE gene was fused with the DNA coding sequence for luciferase and then transfected to Ac2F cells. PPAR-γ agonists will bind and activate the PPAR-γ receptor, and then the activated PPAR-γ will be translocated into the nucleus and bind to PPRE to promote the transcription of relevant genes. The luciferase gene will be simultaneously expressed and the produced light is measured by luminometer. The amount of light will show the ability of PPAR-γ agonists (**1** or **2**) to bind to and activate PPAR-γ receptor. PcDNA (plasmid cloning DNA), PPRE, PPAR-α, or -ϒ plasmids were transfected into Ac2F cells, and treated with **1** or **2** (10, 20, and 40 μM) for 6 h. The cells transfected with PcDNA were employed as the blank group. WY-14643 (10 μM) and rosiglitazone (10 μM) were used as positive controls. The cells transfected with PPAR-α, -ϒ, and PPRE were employed as the control. Despite being less potent than standard PPAR-α and -γ agonists, **2** activated PPAR-α and -γ in a dose-dependent manner, and **1** selectively activated PPAR-γ in a dose-dependent manner.

### 2.4. Molecular Docking of Compound **1** with PPAR-γ

PPAR-γ full agonistic antidiabetic drugs, such as TZDs, are associated with side effects, such as weight gain and fluid retention, which are especially maleficent to diabetic patients. In contrast, partial agonists that exhibit different molecular interactions from TZDs were suggested to retain their hypoglycemic effect and cause negligible weight gain. Metaglidasen, a PPAR-γ partial agonist that does not form the key H-bond with the Tyr^473^ residue as TZDs, is an insulin sensitizer with lower plasma glucose levels but lack side effects, such as weight gain [16,17]. Other synthetic and natural PPAR-γ partial agonists (nTZDpa [14], SR145 [14], BVT.13 [14], and amorfrutin [17]) bind to the PPAR-γ LBD through an H-bond with Ser^342^ of β-sheet and hydrophobic interactions with H3, thereby stabilizing PPAR-γ. The molecular interaction between **1** and PPAR-γ LBD was investigated in silico using human PPAR-γ crystal data (2PRG). According to previous reports, a ligand should first bind to PPAR-γ and stabilize its 3D conformation to activate it. Because **1** activated PPAR-γ, it was reasonably assumed that **1** binds to PPAR-γ. Based on this assumption, we tried to gain an insight about the binding pose of **1** and its potential as a partial agonist. Prior to docking simulation of **1**, we tested simulations of the full agonist rosiglitazone and the partial agonist amorfrutin 1, and compared with reported X-ray crystallographic data. The simulation data matched with the experimental data. Compound **1** was found to occupy the same region as amorfrutin 1 between the H3 and the β-sheet. Such finding indicates that **1** might serve as a PPAR-γ partial agonist, similar to amorfrutin 1 (Figure 4A). The carbonyl group of amorfrutin 1 (blue) formed H-bonds with the amino acid residues Ser^342^ and Glu^343^ (backbone) of the β-sheet. Hydrophobic interactions with Arg^288^ of H3 were indicated (Figure 4B). For compound **1**, H-bonds between 3β-OH and Glu^343^ of the β-sheet were identified, including hydrophobic interactions with H3 (Cys^285^, Arg^288^, and Phe^287^) (Figure 4C). Compound **2** also occupied the region between the H3 and the β-sheet, with a binding affinity score of −7.2 kcal/mol (Appendix A). The binding poses of **1** and amorfrutin 1 were similar to each other, but clearly differed from that of rosiglitazone. The polar moiety of rosiglitazone occupied the region near H11 and H12 by establishing H-bonds with Tyr^473^, His^323^, Ser^289^, and His^449^ [15,19,38] (Figure 4D). The H-bonding with Tyr^473^ (essential for H12 conformation) was proposed to be a master keeper for distinguishing the PPAR-γ full agonist from the partial agonist [15,19,38]. However, partial agonists mainly occupy the region between H3 and β-sheet, and stabilize the conformation of the LBD through an H-bond with Ser^342^ (β-sheet) and hydrophobic interactions with H3 [15,19]. Compound **1** was calculated to form an H-bond with Glu^343^ instead of the adjacent residue, Ser^342^, on the β-sheet. However, the interaction still occurred with H3 and the β-sheet region. Based on docking simulation, compound **1** was identified as a PPAR-γ partial agonist, displaying different molecular interactions from rosiglitazone. Accordingly, compound **1** was investigated for its cellular effects, which are characteristic of PPAR-γ partial agonists.

### 2.5. Compound ***1*** Stimulated the Glucose Uptake in HepG2 Human Liver Cells

In diabetes, PPAR-α and -γ not only ameliorate dyslipidemia, but also hyperglycemia [33]. The potent and effective approach for decreasing blood glucose level involves the stimulation of the liver tissue to absorb blood glucose [39]. The liver plays an important role in regulating glucose homeostasis by controlling the glucose metabolic pathways [40]. PPAR-γ is well-known as a potential target that could improve glucose homeostasis by increasing the glucose and insulin sensitivity of tissues [9]. PPAR-γ activation could also reduce the blood lipid level by absorbing and accumulating lipid into fat cells via adipogenesis stimulation. The formation and growth of new fat cells were considered to be the reason for weight gain or visceral obesity [41]. Further, weight gain was considered to be the key adverse effect of the several antidiabetic drug (TZDs) in diabetic patients [41]. A search for novel PPAR-γ partial agonists with an antidiabetic effect and lacking adverse effects (i.e., adipogenesis and weight gain) would be an attractive strategy. Therefore, glucose uptake and adipocyte differentiation assays were used to discover PPAR-γ partial agonists that stimulate glucose uptake and are accompanied by reduced side effects (adipogenic activity) [42].

To validate the PPAR-γ transactivation effect and glucose uptake activity in HepG2 human liver cells, a luciferase assay and a fluorescent glucose analogue were used. The PcDNA, PPRE, and PPAR-ϒ plasmids were transfected into HepG2 cells using free medium and treated with compounds **1** and **2** for another 6 h. After treatment, the cells were lysed and the luciferase-induced light was measured. Rosiglitazone was used as a positive control. Compounds **1** and **2** both activated PPAR-γ in a concentration-dependent mode in HepG2 cells (Figure 5). The glucose uptake activity was observed using a fluorescence microscope (Figure 6A) and quantified using a fluorescence microplate-reader (Figure 6B). The glucose analogue 2-NBDG (2-(*N*-nitrobenz-2-oxa-1,3-diazo-4-yl) amino)-2-deoxyglucose) was observed as a bright green fluorescence in cells treated with insulin, rosiglitazone, or **1** (20 and 40 μM) (Figure 6A). Compound **1** (20 and 40 μM) was found to promote glucose uptake into HepG2 cells with comparable activities to those of rosiglitazone and insulin, respectively (Figure 6B). Notably, **1** was less potent than **2** in PPAR-v activation (Figure 5), but was more efficient in glucose uptake stimulation (Figure 6B). Although PPAR-γ activation by **1** was markedly weaker than rosiglitazone, the glucose uptake stimulation by **1** was comparable to rosiglitazone at 10 μM (Figure 6B).

### 2.6. Effects of Compound **1** on Lipid Accumulation in 3T3-L1 Preadipocytes

The adipogenesis and lipid-accumulating activity of **1** and **2** were compared with that of rosiglitazone using 3T3-L1 preadipocytes. The 3T3-L1 preadipocytes differentiate and transform to mature adipocytes. The mature adipocytes were stained by Oil Red O dye, and the lipid contents were visualized as red droplets. The Oil Red O staining results showed that **1** and **2** did not induce notable adipogenesis up to a concentration of 40 μM (Figure 7). Compared to the control cells, 3T3-L1 cells treated with **1** and **2** did not show a notable increase in the number of red droplets of Oil Red O-stained mature adipocytes (Figure 7A). In a quantitative assessment, compound **1** showed virtually no extra lipid accumulation compared to the control (Figure 7B). Compared to the PPAR-γ full agonist rosiglitazone (10 μM), the tentative partial agonist **1** showed higher glucose uptake and lower lipid accumulation at concentrations of 20 and 40 μM. Furthermore, the glucose uptake stimulation activity of **1** at 40 μM was comparable to that of insulin at 10 μM. This result suggested that **1** may serve as a lead molecule for the study of antidiabetic agents that lack side effects, such as weight gain and obesity.

## 3. Materials and Methods

### 3.1. General Experimental Procedures

^1^H and 2D NMR spectra were obtained using a Varian INOVA 500 spectrometer while ^13^C NMR spectra were obtained using a Varian UNITY 400 spectrometer. The exact mass of the new compound was analyzed by UHPLC-Q-TOF-MS using UHPLC Agilent 1200 series and 6530 accurate-mass Q-TOF MS with mobile phase comprising water and acetonitrile with 0.1% formic acid. HPLC was performed using an ODS column (YMC-triart C18, 250 × 10.0 mm, i.d. 5 μm) with a Gilson 307 pump and Shodex RI-71 detector.

### 3.2. Materials

3-(4,5-Dimethylthiazol-2-yl)-2,5-diphenyltetrazolium bromide (MTT), WY14643, GW501516, rosiglitazone, 3-Isobutyl-1-methylxanthine (IBMX), dexamethasone, insulin, and Oil Red O were purchased from Sigma-Aldrich (St. Louis, MO, USA). 2-(*N*-(7-Nitrobenz-2-oxa-1,3-diazol-4-yl) amino)-2-deoxyglucose (2-NBDG) was purchased from Invitrogen (Carlsbad, CA, USA).

### 3.3. Extraction and Isolation

*A. flavus* was obtained from the jellyfish *Aurelia aurita* and maintained at the Marine Natural Product Laboratory, Pusan National University. The fungal strain, *A. flavus*, was confirmed using ITS gene sequences (GenBank accession no. KY234273.1) and cultured in 33L YM medium (75% in seawater) for 21 days in a shaker platform at 130 rpm, as previously reported [19]. After extraction and separation by MPLC, 28 fractions were obtained. Compounds **3** (1.74 mg) and **4** (1.56 mg) were obtained from fraction 10 by RP-HPLC (YMC-triart C18, 250 × 10.0 mm, i.d. 5 μm) using 40% methanol/water. Fractions 11, 12, 16, and 17 were separated by RP-HPLC (YMC-triart C18, 250 × 10.0 mm, i.d. 5 μm) with 45%, 48%, 62%, and 66% methanol/water, resulting in compounds **5** (2.87 mg), **6** (5.65 mg), **7** (1.04 mg), and **8** (0.66 mg). Compounds **1** (1.36 mg) and **2** (2.4 mg) were isolated from fractions 15 and 16 by RP-HPLC (YMC-triart C18, 250 × 10.0 mm, i.d. 5 μm) using 58% methanol/water with retention times of 125 and 185 min, respectively. Compounds **9** (2.7 mg), **10** (1.5 mg), and **11** (0.95 mg) were obtained from fraction 21 via elution with 76% methanol/water.

Asperflagin, oleana-11,13(18)-diene-3β,15α,22β,24,30-pentol (**1**): 1.36 mg. [α]^25^_D_ -1.6 (*c* = 0.6, MeOH); UV λ_max_ (nm) 210, 250. HRESIMS *m/z* 533.3484 [M + HCOO]^−^ (∆ +1.1 ppm/+0.6 mmu). Q-TOF MS is widely employed for the analysis of saponins. The peak at *m/z* 533.3484 shows a solvent (formic acid) adduct to [M-H]^−^ The peak ratio of [M-H]^−^ to [M + HCOO]^−^ was determined using the sugar chains [43]. However, aglycones, such as compound **1**, displayed only [M + HCOO]^−^, as demonstrated previously [43]. The ^1^H NMR (500 MHz, CD_3_OD) and ^13^C NMR data (400 MHz, CD_3_OD) are presented in Table 1.

### 3.4. Computational Methods

The PDB code, 2PRG, of PPAR-γ was downloaded from the Protein Data Bank. The protein 2PRG and ligands were prepared as previously reported [19]. Docking calculations were performed using AutoDock Vina 1.1.2 (The Scripps Research Institute, La Jolla, CA, USA). The protein–ligand interaction was analyzed and visualized using Discovery Studio 4.5 (NeoTrident Technology Ltd., Beijing, China).

### 3.5. Cell Culture and Cell Viability Assay

Rat liver (Ac2F), preadipocytes (3T3-L1), and human hepatoma (HepG2) cells were purchased from the American Type Culture Collection (ATCC, Rockville, MD, USA), and incubated in Dulbecco’s modified Eagle’s medium (Hyclone, Logan, UT, USA) supplemented with 10% fetal bovine serum (FBS) (Gibco-BRL, NY, USA) and 1% penicillin/streptomycin at 37 °C in a humidified 5% CO_2_. A total of 1 × 10^4^ cells/well Ac2F cells were seeded in 96-well plates for 15 h and treated with compounds **1**–**11** (10 and 40 μM) for 12 h in free medium. After treatment, 20 μL 3-(4, 5-Dimethylthiazol-2-yl)-2, 5-diphenyltetrazolium bromide (MTT) (0.5 mg/mL) was added to the 96-well plates, which were incubated in the dark for 4 h. After the supernatants were removed from the wells of the 96-well plates, 150 μL DMSO was added to dissolve the formazan crystals. After shaking, the plates were analyzed using a microplate reader (Elx 800, Bio-Tek, Winooski, VT, USA) at 490 nm.

### 3.6. Luciferase Assay

Ac2F or HepG2 cells were seeded in 48-well plates and incubated until 80% confluence was achieved. Thereafter, these cells were transfected with TK-PPRE × 3-luciferase reporter plasmid (1 µg/well) and pcDNA3 (0.1 µg/well) or PPAR-α (0.1 µg/well), β/δ (0.1 µg/well), or PPAR-γ1 (0.1 µg/well) for 4 h in medium (as described in our previous report [44]). After transfection, the free medium was changed to complete medium (DMEM/10% FBS) for overnight culture. Compounds **1**–**11** (10 µM or 10, 20, and 40 µM), PPAR-α agonist WY14643 (10 µM), the PPAR- β/δ agonist, GW501516 (10 µM), and the PPAR-γ agonist, rosiglitazone (10 µM), were added to the plates containing transfected cells for 6 h in free medium. After drug treatment, the cells were lysed and analyzed by ONE-Glo™ Luciferase Assay System (Promega, Madison, WI, USA), and the data were collected using a GloMax^®^-Multi Microplate Multimode Reader (Promega Co., Madison, WI, USA).

### 3.7. Adipocyte Differentiation Assay

A total of 5 × 10^4^ cells/well 3T3-L1 cells were seeded in 48-well plates for 48 h to achieve confluence. After confluency, the cells were treated with compound **1** (10, 20, and 40 µM), **2** (10, 20, and 40 µM), and rosiglitazone (10 µM) in complete medium (DMEM/10% FBS) containing 0.5 mM 3-Isobutyl-1-methylxanthine (IBMX), 1 μM dexamethasone, and 1 μg/mL insulin for 48 h. Thereafter, the cells were treated with compound **1** (10, 20, and 40 µM), **2** (10, 20, and 40 µM) and rosiglitazone (10 µM) in complete medium containing 1 μg/mL insulin for 48 h. Thereafter, the medium was replaced with complete medium and the samples were incubated for another 48 h. After drug treatment, the cells were washed with PBS and fixed with 70% ethanol for 30 min. The fixed cells were then stained using Oil Red O for 1 h. The stained cells were washed using dH_2_O and analyzed using an Optinity (Gyeonggi-Do, Korea) microscope. 100% isopropanol was used to elute Oil Red O, and the iMark Microplate Absorbance Reader (Bio-Rad Laboratories, Hercules, CA, USA) was used to measure the OD value at 520 nm.

### 3.8. Glucose Uptake Assay

HepG2 cells were seeded in 96-well plates and incubated for 24 h. The complete medium was changed to a medium without glucose and the cells were incubated for another 24 h. Thereafter, compound **1** (10, 20, and 40 µM), **2** (10, 20, and 40 µM), insulin (10 µM), and rosiglitazone (10 µM) with 2-NBDG (50 µM) were administered to cells for 1 h. After drug treatment, the cells were washed with PBS, and the fluorescence was measured using a GloMax^®^-Multi Microplate Multimode Reader (Promega Co., Madison, WI, USA) at an excitation wavelength of 460 nm and emission wavelength of 540 nm.

HepG2 cells were seeded in confocal dishes and cultured for 24 h. After the medium was changed to medium without glucose, the cells were incubated for 24 h. Compound **1** (10, 20, and 40 µM), **2** (10, 20, and 40 µM), insulin (10 µM), and rosiglitazone (10 µM) with or without 2-NBDG (50 µM) were administered to the cells for 1 h. The fluorescence images of cells were obtained using a ZEISS LSM 800 confocal microscope (Oberkochen, Baden-Württemberg, Germany).

### 3.9. Statistical Analysis

GraphPad Prism 5 (San Diego, CA, USA) software was used for data analysis. The data are presented as the mean ± standard error of the mean. The significant difference between group was determined using one-way analysis of variance and Tukey’s HSD-post hoc test. * *p* < 0.05, ** *p* < 0.01, and *** *p* < 0.001 were considered to indicate a significant difference.

## 4. Conclusions

In summary, based on an investigation of the PPAR agonistic components from the jellyfish-derived fungus, *A. flavus*, a new triterpenoid **1** was identified as a potential PPAR-γ partial agonist. PPAR-γ activation by a full agonistic antidiabetic drug is reported to induce side effects, such as weight gain and obesity, which are undesirable to diabetic patients. However, partial agonists, which have a molecular interaction that is distinct from that of full agonists, could still retain the antidiabetic effect without triggering side effects. The typical difference in ligand–receptor interactions between PPAR-γ full and partial agonists was already defined at the molecular level by X-ray crystallographic studies. Therefore, the molecular interaction between **1** and human PPAR-γ LBD was investigated by docking simulation. Molecular docking simulation suggested that **1** is located between the H3 and the β-sheet regions of human PPAR-γ LBD in a similar manner as other partial agonists that lack side effects. In a cell-based assay, **1** displayed better glucose uptake in HepG2 human liver cells at 20 and 40 μM than rosiglitazone, but induced markedly lower adipogenesis and lipid accumulation in 3T3-L1 preadipocytes. These results suggest that **1** might function as a typical PPAR-γ partial agonist and may be utilized for further studies on more advanced antidiabetic agents.

## Figures and Tables

**Figure 1 marinedrugs-20-00203-f001:**
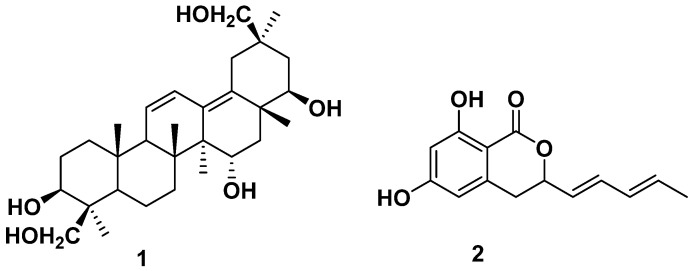
Structures of compounds **1** and **2** isolated from the jellyfish-derived fungus *Aspergillus flavus*.

**Figure 2 marinedrugs-20-00203-f002:**
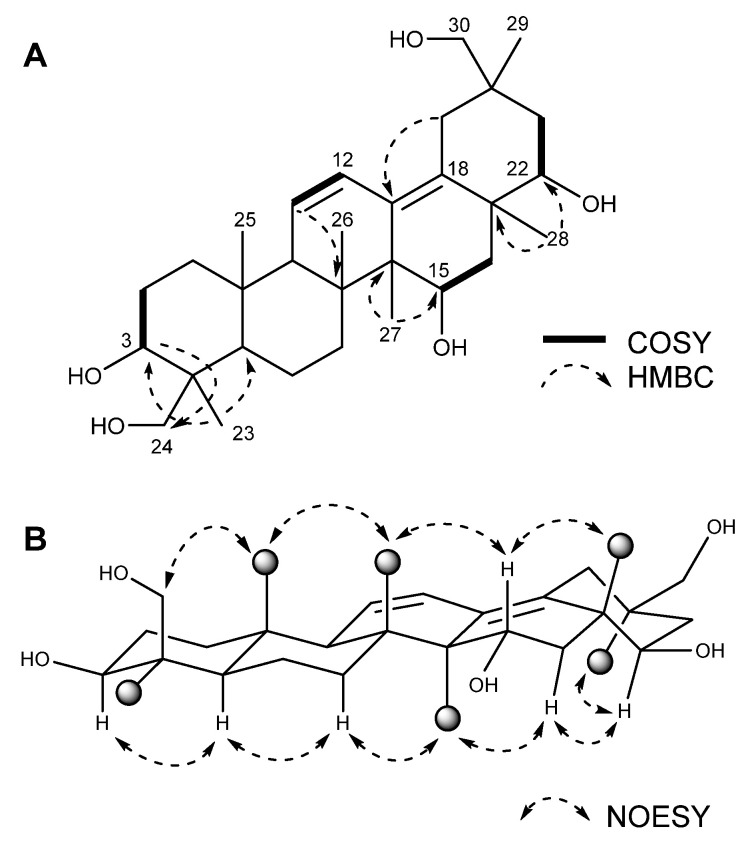
Key HMBC, COSY, and NOESY correlation data for compound **1**. (**A**) Key COSY and HMBC correlations. (**B**) Key NOESY correlations. The dark sphere in the NOESY data represents a methyl group.

**Figure 3 marinedrugs-20-00203-f003:**
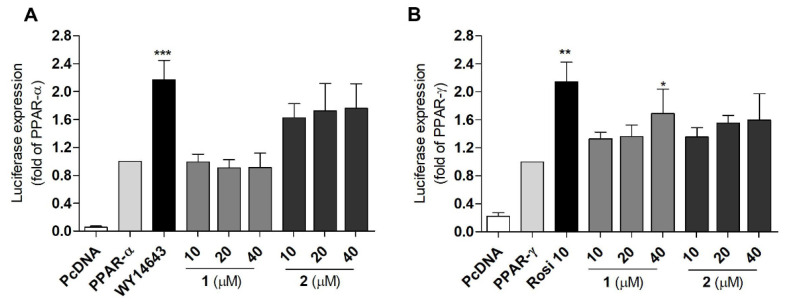
Transactivation effects of PPAR-α and -ϒ in Ac2F cells treated with **1** and **2**. (**A**) The transactivation effects of PPAR-α following treatment with 10, 20, and 40 μM of **1** and **2**. (**B**) The transactivation effects of PPAR-ϒ following treatment with 10, 20, and 40 μM of **1** and **2**. WY-14643 (10 μM) and rosiglitazone (10 μM) were used as positive controls. The cells transfected with PcDNA were employed as the blank group. The cells transfected with PPAR-α, -ϒ, and PPRE were employed as the control. WY-14643 (10 μM) and rosiglitazone (10 μM) were used as positive controls. * *p* < 0.05 ** *p* < 0.01, *** *p* < 0.001 vs. PPAR-α or -ϒ.

**Figure 4 marinedrugs-20-00203-f004:**
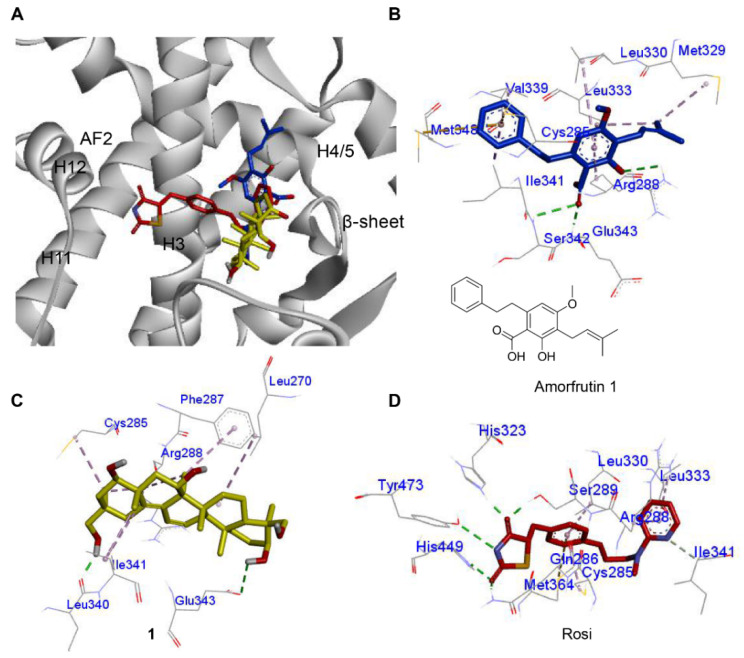
(**A**) Comparison of the individual docking poses of amorfrutin **1** (blue), **1** (yellow), and rosiglitazone (red) into PPAR-γ LBD (PDB: 2PRG). (**B**) The binding interactions of amorfrutin 1 (−8.2 kcal/mol). (**C**) The binding interactions of **1** (−8.7 kcal/mol). (**D**) The binding interactions of rosiglitazone (−8.5 kcal/mol). An H-bond is depicted by a green dashed line; a hydrophobic interaction is depicted by a pink dashed line.

**Figure 5 marinedrugs-20-00203-f005:**
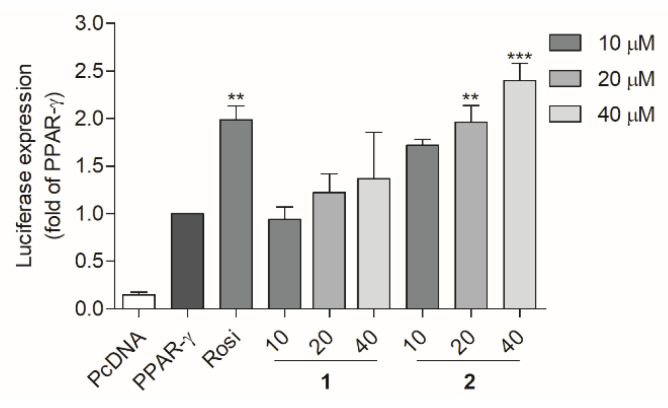
PPAR-ϒ transactivation effects of **1** and **2** in HepG2 cells. Rosiglitazone was used as a positive control. The cells transfected with PcDNA were employed as the blank group. Cells transfected with PPAR-ϒ and PPRE were employed as the control. ** *p* < 0.01, *** *p* < 0.001 vs. PPAR-ϒ.

**Figure 6 marinedrugs-20-00203-f006:**
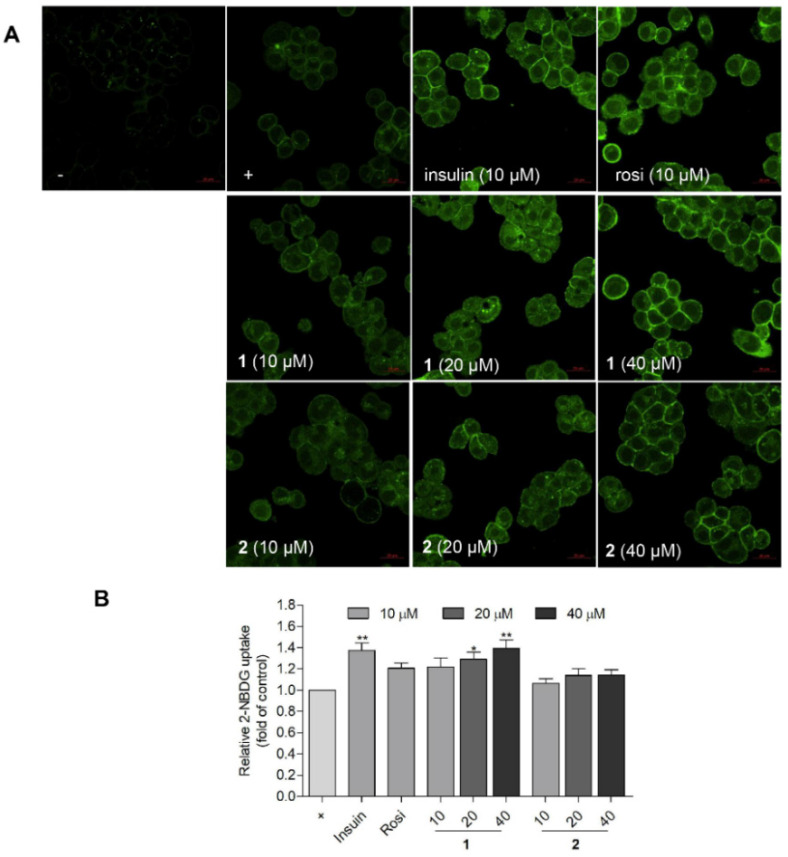
Glucose uptake stimulation activities of **1** and **2** in HepG2 cells. (**A**) HepG2 cells were treated with insulin (10 μM), rosiglitazone (10 μM), **1** (10, 20, and 40 μM), or **2** (10, 20, and 40 μM), in addition to 2-NBDG (50 μM) for 1 h and observed using a fluorescence microscope. (−): No 2-NBDG. (+): only 2-NBDG treatment. (**B**) Quantitative presentation of the glucose uptake stimulation activity. The fluorescent signals were analyzed by detecting the fluorescence at Ex/Em = 460/540 nm. Insulin and rosiglitazone were used as positive controls. (+): 2-NBDG treatment without drug. * *p* < 0.05, ** *p* < 0.01 vs. control.

**Figure 7 marinedrugs-20-00203-f007:**
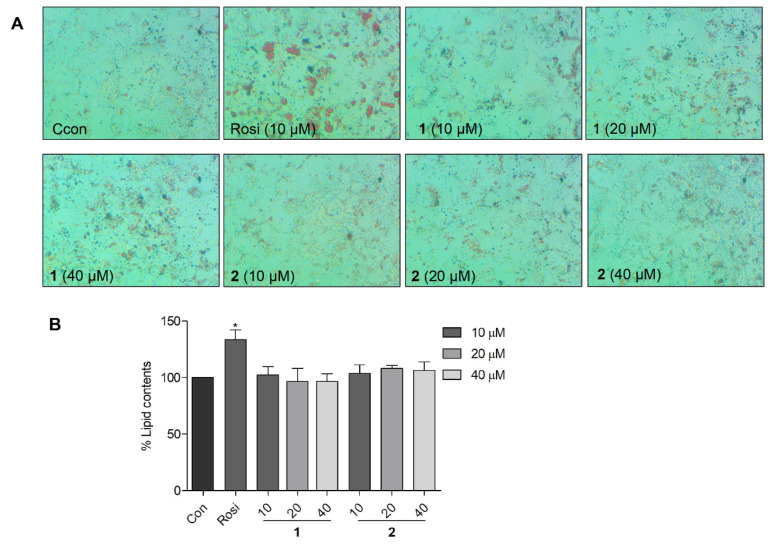
Adipogenic activity of compounds **1** and **2** in 3T3-L1 cells. (**A**) Morphological assessment of adipocyte differentiation after the treatment of 3T3-L1 cells with **1** (10, 20, and 40 μM) or **2** (10, 20, and 40 μM). Red droplets represent mature adipocytes stained using the Oil Red O dye. (**B**) Quantitative presentation of the lipid content. * *p* < 0.05 vs. control.

**Table 1 marinedrugs-20-00203-t001:** ^1^H (500 MHz) and ^13^C (100 MHz) NMR data of compound **1** (CD_3_OD).

No	δ_H_, Mult (J in Hz)	δ_C_	No	δ_H_, Mult (J in Hz)	δ_C_
1	1.95 (overlapped) 1.66 (overlapped)	39.1	16	1.91 (overlapped) 1.60 (overlapped)	43.9
2	1.85 (overlapped) 1.76 (overlapped)	28.3	17		43.6
3	3.38, dd (14.0, 4.9)	81.2	18		137.2
4		43.6	19	2.36, d (14.7) 1.85, d (14.7)	32.9
5	0.95 (overlapped)	56.6	20		38.2
6	1.73, m 1.46, m	20.1	21	1.62 (overlapped) 1.36 (overlapped)	38.8
7	1.63, m	36.9	22	3.44, dd (10.0, 5.0)	77.5
8		42.7	23	1.22, s	23.0
9	1.94 (overlapped)	55.6	24	4.09, d (11.0) 3.42, d (11.0)	65.0
10		37.7	25	0.92, s	18.8
11	5.64, d (10.5) *	127.6	26	0.77, s	17.8
12	6.36, dd (10.5, 3.0) *	127.4	27	1.00, s	14.9
13		136.8	28	1.09, s	19.1
14		48.8	29	0.80, s	20.9
15	4.03, dd (12.5, 4.1)	66.7	30	3.28, s	73.7

* The assignment of H-11 and H-12 was based on the NOE correlations between H-11 and H-9, and between H-12 and H-19a. The deceptive splitting pattern of H-11 (d) and H-12 (dd) may be explained by examination of the stable 3D conformation of **1**. The dihedral angle between H-11 and H-9 is close to 90°, which yields a negligible scalar coupling between them. Whilst, a long-range W (zig-zag) coupling (3 Hz) between H-12 and H-9 would be possible.

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
