# Peer review of "A New Fungal Triterpene from the Fungus Aspergillus flavus Stimulates Glucose Uptake without Fat Accumulation"

_marinedrugs, 2022, doi:10.3390/md20030203_

Round 1

Reviewer 1 Report

The manuscript entitled "A new fungal triterpene stimulates glucose uptake without fat accumulation" presents results dealing with the extraction,  identification and biological activity of fungal compounds from which one was described as new products. Overall, the manuscript is interesting and well written. I recommend its possible publication taking into account the following minor remarks:

  • I think that the manuscript title should contain the name of the starting material from which the discussed compounds were extracted. Its scientific name should also added in the key words list.
  • A suitable legend should be added to the MS and NMR spectra indicated in the supplementary file.  Ions corresponding to the signals observed in the MS spectrum should be indicated. Suitable legend should be added to the signals and cross peaks observed in the 1D and 2D NMR spectra in order to indicated the corresponding C and/or H atoms. 

Author Response

Please see the file attachment.

Reviewer 2 Report

The paper by Li and co-workers reports the discovery of a new bioactive triterpene. 

In my opinion, from a general point of view, this papers deserves publication in Marine Drugs but it needs to be improved in some points:

1.- The introduction needs to be clearer. For non-specialist in the development of antidiabetics, as is my case, the section related with the molecular mechanisms involved in the PPARs mediated regulation of glucose metabolism have to be improved (3rd paragraph). I also suggest to include a scheme.

2.- Structure elucidation of 1.

a) Multiplicity of protons is necessary for most protons. I suggest to acquire selective 1D-TOCSY spectra to achieve this goal. This values will help to secure the proposed relative configuration

b) Why the chemical shift of H3 (3.38) indicates and alpha orientation ?

3.- PPAR Transactivation 

a) Could you further explain the rationale behind the luciferase assay shown in figure 3 ?

b) Please, explain with more details the results shown in figure 3 in the main text ?

4.- Molecular docking

a) It is absolutely necessary that the authors confirm that compound 1 really binds PPAR-gamma. In vitro protein-ligand binding studies should be undertaken. Otherwise, section 2.4 is not solid enough. 

b) Authors should check their methodology using other ligands with known results for the same protein. The absolute binding affinity energy obtained from this kind of calculations (line 162) are useless without relative references.

5.- With regard to sections 2.5 & 2.6 I have to say that I am not a pharmacologist (just study pharmacy). The results and the rationale behind the assays needs to be explained clearer for the general reader (mainly chemist) of this journal. 

In summary, the results and the mechanism of action need to be clarified. In particular, it is crucial to confirm if molecule 1 binds to PPAR because this is the main claimed goal of the discovery. Otherwise, there are lots of molecules with anti diabetic activity and the paper would not have a great impact in the scientific community. 

Reviewer 3 Report

The manuscript entitled (A new fungal triterpene stimulates glucose uptake without fat accumulation) by li et al. describe a new triterpene from A. flavus. The structure was elucidated by various spectroscopic tools and tested for its agonist towards PPAR-g. The manuscript could be accepted after covering the following issues.

1- The compound was isolated from Aspergillus flavus, so the authors should mention the active constituents previously isolated from the fungus and Jellyfish Aurelia aurita as well as, the compounds that have this activity from the fungus or Jellyfish Aurelia aurita.

2- Structural elucidation of compound 1 should be discussed in deep (H, C, COSY and HMBC).

3- Ten known compounds were identified by which methods, the author should mention the identification methods of the known compounds.

4- To increase the citation of this paper, please add 1D NMR of Known compounds in the supplementary material.

5- Structures of compound 2 should be drawn with compound 1. Other compounds` structures should add in supplementary.

6- Compounds 1 and 2 were isolated by HPLC, please add Rt for each compound in the experimental section.

7- use the trivial name of compound 1 in the abstract, also add it in the keywords.

8- is this class of triterpenes commonly encountered in fungi especially Aspergillus species. Discuss the possible biosynthetic pathway.

9-The alpha-D and IR for compound 1 should be added.

Round 2

Reviewer 2 Report

In my opinion, the new version of the manuscript is clearer for the reader than the previous one. Still, from my point of view there a few minor points that need to be included. Basically, there are explanations for all my previous concerns on the response letter but not on the manuscript.

1.- Answer to comment 2b, should be included in the manuscript, not only a new reference.

2.- Answer to comment 4a should be included in the manuscript.

Author Response

We appreciate the very valuable comments. We revised the manuscript as advised.

1.- Answer to comment 2b, should be included in the manuscript, not only a new reference.

--- We included a sentence as suggested as follows.

"The chemical shift of H-3 (δ 3.38, The chemical shift of ß-oriented H-3 would be higher than 3.6 ppm) and the coupling constant of 14.0 Hz between 2-Hax and 3-Hax indicated an α-orientation of H-3 [22,23], and the NOE correlations (Figure 2B)."

2.- Answer to comment 4a should be included in the manuscript.

--- Following sentences were added as advised.

"According to previous reports, a ligand should first bind to PPAR-γ and stabilize its 3D conformation to activate it. Because 1 activated PPAR-γ, it was reasonably assumed that 1 binds to PPAR-γ. Based on this assumption, we tried to gain an insight about the binding pose of 1 and its potential as a partial agonist."

Reviewer 3 Report

No comments

Author Response

Thank you for your considerate review.

There was no comment by the reviewer and the evaluations were marked 'yes'.